# The role of Notch signaling in endometrial mesenchymal stromal/stem-like cells maintenance

Sisi Zhang[1,2,3], Rachel W. S. Chan [1,2,3✉], Ernest H. Y. Ng [1,2] & William S. B. Yeung [1,2✉]

Human endometrium undergoes cycles of regeneration in women of reproductive age. The endometrial mesenchymal stromal/stem cells (eMSC) contribute to this process. Notch signaling is essential for homeostasis of somatic stem cells. However, its role in eMSC remains unclear. We show with gain- and loss-of-function experiments that activation of Notch signaling promotes eMSC maintenance, while inhibition induces opposite effect. The activation of Notch signaling better maintains eMSC in a quiescent state. However, these quiescent eMSC can re-enter the cell cycle depending on the Notch and Wnt activities in the microenvironment, suggesting a crosstalk between the two signaling pathways. We further show that the Notch signaling is involved in endometrial remodeling event in a mouse menstrual-like model. Suppression of Notch signaling reduces the proliferation of Notch1$^+$ label-retaining stromal cells and delays endometrial repair. Our data demonstrate the importance of Notch signaling in regulating the endometrial stem/progenitor cells in vitro and in vivo.

[1] Department of Obstetrics and Gynaecology, LKS Faculty of Medicine, The University of Hong Kong, Hong Kong SAR 999077, China. [2] Shenzhen Key Laboratory of Fertility Regulation, The University of Hong Kong Shenzhen Hospital, Shenzhen 518000, China. [3]These authors contributed equally: Sisi Zhang, Rachel W.S. Chan. ✉email: rwschan@hku.hk; wsbyeung@hku.hk

Human endometrium undergoes cyclical proliferation, differentiation and shedding during women's reproductive years[1]. Regeneration of the endometrium after menses is an essential component of a menstrual cycle in humans and primates[2,3]. Endometrial stem/progenitor cells are vital in tissue regeneration after menstruation[4]. Schwab and Gargett isolated a rare population of human endometrial mesenchymal stromal/stem-like cells (eMSC) based on co-expression of two perivascular markers: CD140b and CD146[5]. These cells showed somatic stem cell properties and phenotypes similar to other mesenchymal stem/stromal cells[6,7].

Notch is a highly conserved signaling pathway vital in tissue homeostasis and maintenance of somatic stem cells[8]. Specific ligand-receptor interactions activate the Notch signaling, leading to a series of proteolytic events and release of Notch intracellular domain (NICD), which is then translocated to the nucleus where it provokes transcription of Notch target genes such as *HEY1*, *HEY2*, and *HES1*[9]. Several Notch family members are expressed in human endometrium[10]. Dysregulation of Notch signaling molecules, Notch1, DLL1, and JAG1 has been observed in the endometrium from infertile women, suggesting an association of Notch signaling with infertility[11]. Notch signaling has been implicated in endometrial remodeling events such as decidualization and embryo implantation[10], and uterine-specific Notch1-knockout mice exhibit decidualization defect[12]. Decreased Notch signaling is also associated with endometriosis[13]. Gene expression profiling reveals increased activation of Notch signals in eMSC when compared with endometrial stromal fibroblasts[14]. Our group recently demonstrated that hypoxia regulated self-renewal of eMSC through Notch signaling[15]. However, the role of Notch signaling on proliferation and maintenance of eMSC remains largely undefined.

WNT/β-catenin pathway is another signaling pathway known to be involved in self-renewal and maintenance of somatic stem cells[16]. Myometrial cells can facilitate self-renewal of eMSC through the WNT5A/β-catenin signaling[17]. Moreover, soluble secretory factors from endometrial niche cells at menstruation modulate the biological activities of eMSC via activation of the WNT/β-catenin signaling[18,19]. Accumulating evidence suggests a crosstalk between the Notch and the WNT/β-catenin signaling in somatic stem cells[20,21]. We postulate that such interaction exists in eMSC.

To explore the role of Notch signaling on the biological activities of eMSC, we examined the expression of Notch activities in different subpopulations of human endometrial stromal cells and determined the interaction of Notch signaling with WNT/β-catenin signaling in eMSC in vitro and in vivo. The functional significance of the Notch signals in endometrial regeneration was studied in a mouse model simulating menstruation with the use of label retaining cell technique for identification of mouse endometrial stem/progenitor cells in vivo.

## Results

### Activation of Notch signaling maintains phenotypic expression of eMSC

To determine the role of Notch signaling on maintenance of eMSC, we first compared the endogenous level of Notch target genes in unfractionated stromal cells and eMSC. *HES-1* (Fig. 1a) and *HEY-L* (Fig. 1b) were abundantly expressed in the eMSC than in the unfractionated stromal cells. The expression of *HEY-1* (Fig. 1c) and *HEY-2* (Fig. 1d) was similar between unfractionated stromal cells and eMSC.

Gain- and loss-of-function approaches were then used to assess the role of Notch signaling on phenotypic expression of eMSC. The percentage of CD140b$^+$CD146$^+$ cells was significantly higher for eMSC cultured on plates coated with a Notch ligand JAG1 than the control eMSC (Fig. 1e), and treatment with DAPT to inhibit the γ-secretase complex abolished the difference (Fig. 1e). Immunofluorescence staining showed an increase in nuclear accumulation of NICD (CD140b$^+$CD146$^+$NICD$^+$) in the eMSC cultured on the JAG1-coated plates, indicating activation of the Notch signaling (Fig. 1f and Supplementary Fig. 1). Concordantly, DAPT treatment reduced the proportion of NICD$^+$ eMSC (Fig. 1f and Supplementary Fig. 1). The biological actions of JAG1 on activation of Notch signaling was confirmed by western blotting. Treatment with recombinant JAG1 enhanced the protein expression of NICD, HES-1, and HEY-2 in eMSC when compared to DMSO control (Fig. 1g). The expression of these Notch target proteins was reduced after treatment with DAPT (Fig. 1g).

### Notch1 mediates the maintenance effect of JAG1 on eMSC

Next, the Notch1 receptor was evaluated. Although the number of amplification cycles of *Notch1* was similar between eMSC and unfractionated stromal cells (Fig. 2a), western blotting (Fig. 2b) and immunofluorescence staining (Fig. 2c) revealed that the Notch1 protein expression was significantly higher in the eMSC than the unfractionated stromal cells. Triple immunofluorescence staining confirmed the co-expression of CD140b, CD146, and Notch1 in 70% of the freshly isolated eMSC (Fig. 2d).

To study the action of JAG1–Notch1 interaction on eMSC, we evaluated the binding of fluorescence labeled JAG1 after knockdown of Notch1. The fluorescence intensity of bound JAG1 on eMSC was reduced after transfection with Notch1 siRNA (Fig. 2f). Knockdown of Notch1 also abolished the maintenance effect of JAG1 on eMSC phenotypic expression (Fig. 2g).

### Activation of Notch signaling maintains eMSC quiescence in vitro

Notch signaling is required for the maintenance of quiescence in somatic stem cells[22,23]. Therefore, we assessed the proliferation activity of eMSC after activation and inhibition of the Notch pathway. The rate of proliferation in eMSC was significantly lower after culture on JAG1 coated plate than on fibronectin coated plate, while no difference was detected between the DAPT treated eMSC and the control group (Fig. 3a). The mRNA (Fig. 3b) and protein (Fig. 3c) expression of the proliferation marker Ki67 in eMSC were also lower after culture on the JAG1 coated plate than on the fibronectin coated plate. We also evaluated the extent of confluence of eMSC after culture for 7 days. Phase-contrast images collected during culture showed that the eMSC were not in direct contact, suggesting that the decrease in proliferation was not related to contact inhibition (Supplementary Fig. 2a). Moreover, we determined the toxicity of DAPT in eMSC. Our results revealed that only high concentration of DAPT (10 μM) could inhibit the proliferation of eMSC (Supplementary Fig. 2b). The DAPT concentration (1.25 μM) used in this study did not induce apoptosis in eMSC as the treatment had no effect on the mRNA expression of the proapoptotic *BCL-2* (Supplementary Fig. 2c) and the antiapoptotic *BAX* (Supplementary Fig. 2d).

To investigate whether the reduction in proliferation activity was related to quiescence of the stem cells, we analyzed the cell cycle status of eMSC. Figure 3d showed the cell cycling status of eMSC with Notch activation or inhibition determined by flow cytometry analysis. JAG1 induced more eMSC at the G0 state and decreased the proportion of cells in the G1 state (Fig. 3e). Blocking the Notch signaling by DAPT reversed the effect (Fig. 3e). Consistent with the cell cycle analysis, activation of the Notch signaling pathway upregulated the mRNA expression of *CDKN1A*, *CCND1* and downregulated that of *CCNA2*, *CCNE2* and *GOS2* (Fig. 3f). However, DAPT showed little effect on the expression of these cell cycle related genes (Fig. 3f). These findings suggest that activation of Notch signaling can better maintain eMSC in a quiescent state.

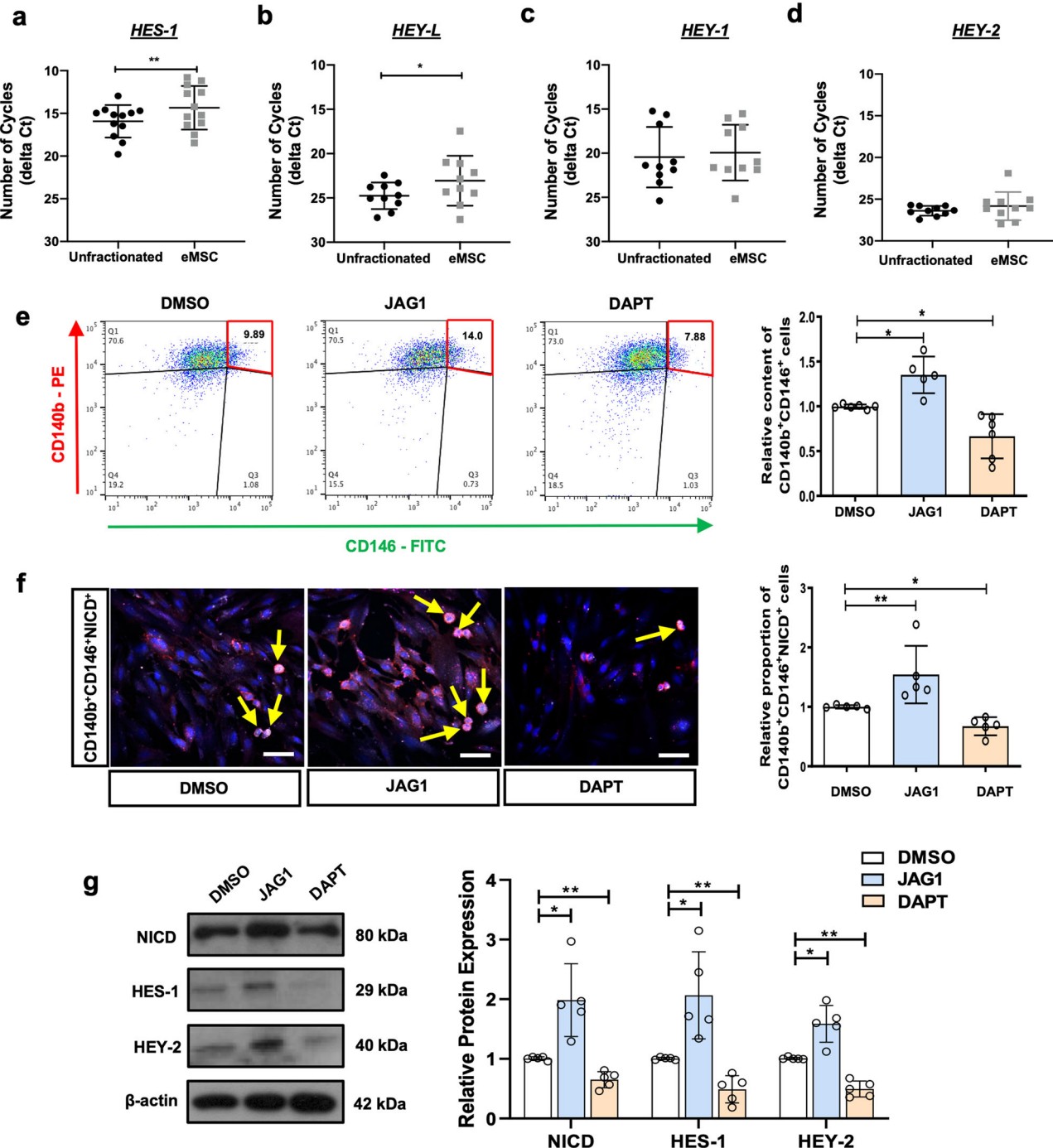

**Fig. 1 Activation of Notch signaling maintains phenotypic expression of eMSC. a–d** Number of amplification cycles (inverted *Y*-axis) for detection of Notch target gene in unfractionated stromal cells and eMSC ($n = 10$–12). **e** Representative dual parameters fluorescence dot plots and histograms showing relative content of CD140b+CD146+ cells after Notch signaling activation and inhibition in eMSC ($n = 6$). **f** Representative immunofluorescence images and quantitative analysis of CD146+CD140b+NICD+ (yellow arrows) cells after Notch signals activation and inhibition ($n = 5$). Scale bar: 50 μm. **g** Representative western blotting images and semi-quantitative analysis of Notch signals protein expression ($n = 5$). Results are presented as mean ± SD; *$P < 0.05$; **$P < 0.01$. Statistical analysis was performed using a two-tailed paired Student's t test for two group comparison. One-way ANOVA followed by Tukey's test for multiple group comparison. eMSC endometrial mesenchymal stem-like cells, NICD notch intracellular domain.

It has been reported that WNT ligands promote the formation of eMSC colonies[17]. To determine whether the JAG1-induced quiescent state of eMSC was reversible, the action of JAG1 in the presence of WNT3A or WNT5A on colony formation was evaluated (Fig. 3g). Indeed, the inhibitive effect of JAG1 on the clonogenicity of eMSC was reversed upon treatment with WNT3A or WNT5A (Fig. 3g). Moreover, DAPT showed little effect on the colony formation of eMSC, and treatment with WNT3A or WNT5A increased the clonogenicity of eMSC (Fig. 3g). Taken together, these findings suggested that the JAG1-induced quiescent eMSC could re-enter into the cell cycle upon WNT activation.

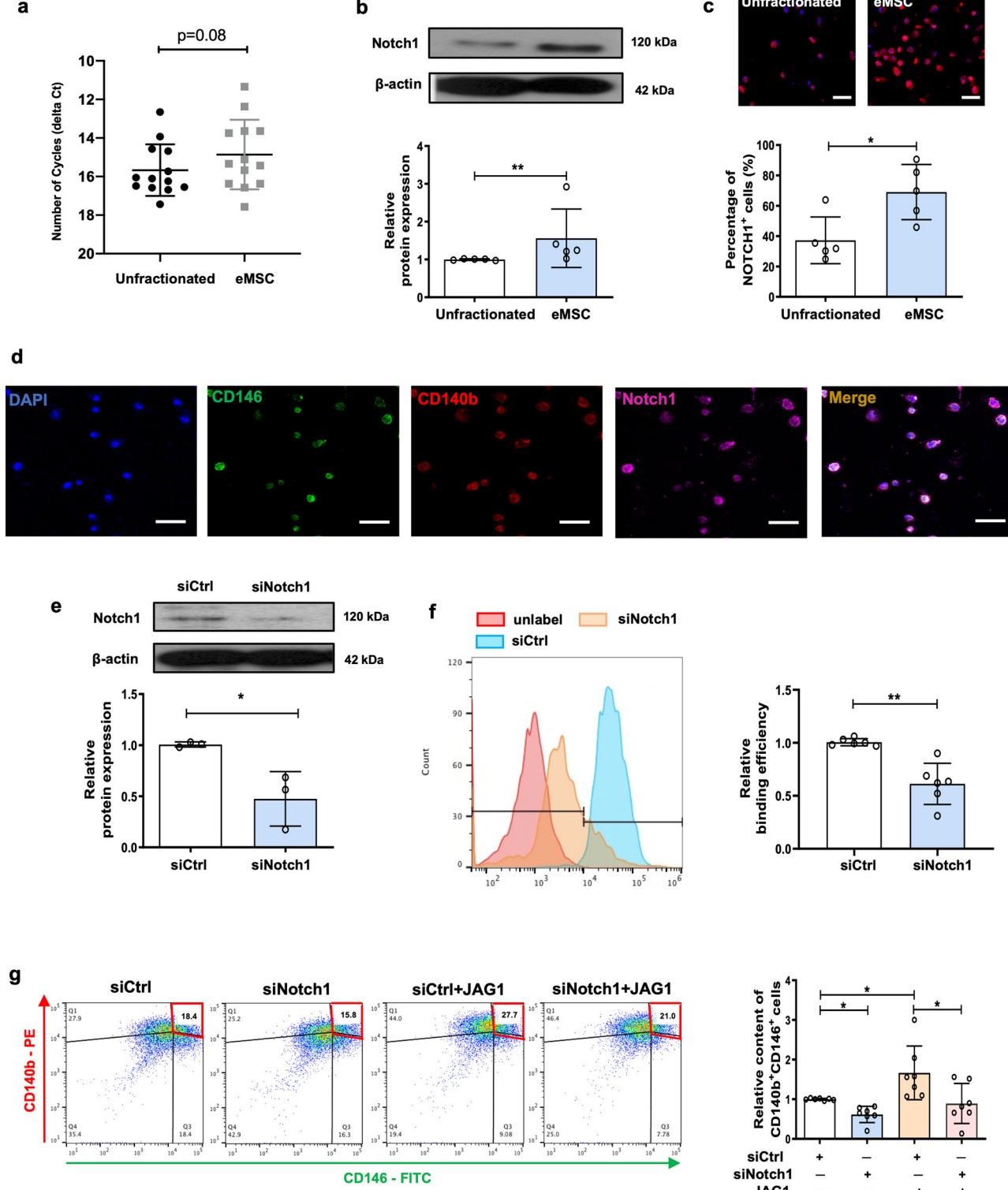

**Fig. 2 The role of Notch1 receptor on eMSC maintenance. a** Number of amplification cycles (inverted *Y*-axis) for detection of *Notch1* receptor in unfractionated stromal cells and eMSC ($n = 13$). **b** Representative western blotting images and semi-quantitative analysis of Notch1 protein expression in unfractionated stromal cells and eMSC ($n = 5$). **c** Representative immunofluorescence images and quantitative analysis of Notch1 in unfractionated stromal cells and eMSC ($n = 5$), Scale bar: 50 μm. **d** Representative immunofluorescence images showing the co-expression of CD140b (red), CD146 (green) and Notch1 (pink) on freshly isolated eMSC ($n = 5$), Scale bar: 50 μm. **e** Representative western blotting images and semi-quantitative analysis of Notch1 protein expression on eMSC ($n = 3$). **f** Representative flow histograms and quantitative analysis of binding efficiency between JAG1 and Notch1 receptor in eMSC ($n = 6$). **g** Representative dual parameter fluorescence dot plot and histogram showing relative percentage of CD140b+CD146+ cells in eMSC upon knockdown of Notch1 with or without JAG1 treatment ($n = 7$). Results are presented as mean ± SD; *$P < 0.05$; **$P < 0.01$. Statistical analysis was performed using a two-tailed paired Student's t test for two group comparison. One-way ANOVA followed by Tukey's test for multiple group comparison. eMSC endometrial mesenchymal stem-like cells, siNotch1 siRNA to Notch1, siCtrl scrambled control siRNA.

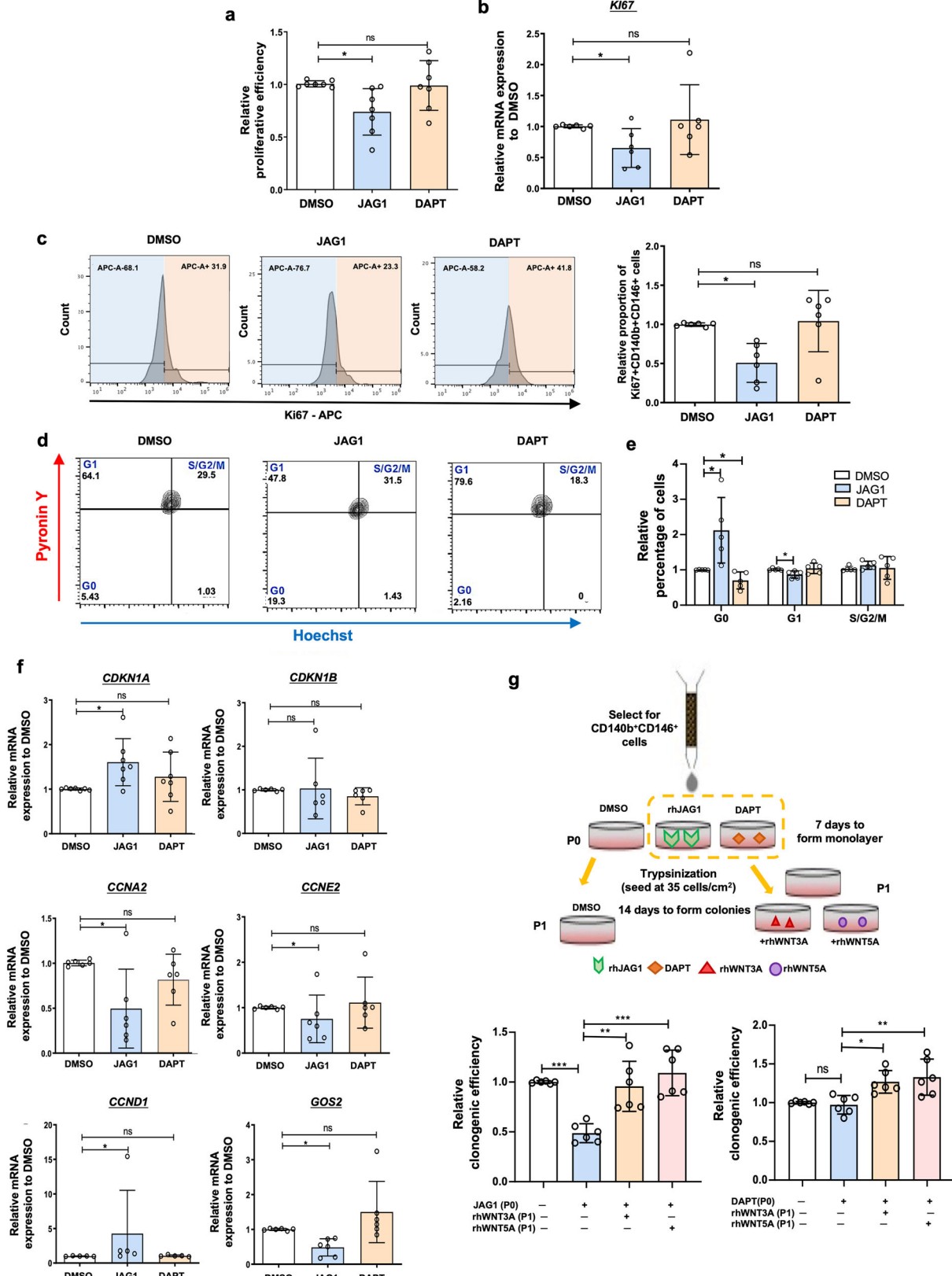

## Interaction of Notch and WNT/β-catenin pathway in eMSC

*Action of Notch activity on active β-catenin.* Crosstalk between the Wnt and the Notch pathway has been reported in different systems[21]. Our above results indicated that the JAG1-induced quiescent eMSC could be reactivated by Wnt activation, suggesting a crosstalk between these two signaling pathways. Therefore, we

further investigated the interaction of the Notch and the WNT/β-catenin pathway in eMSC. Immunofluorescence staining (Supplementary Fig. 3) and western blotting (Fig. 4b) showed that the expression of active β-catenin in eMSC increased after activation and suppressed upon inhibition of Notch signaling. The canonical Wnt signaling is largely mediated by the TCF/LEF family of

**Fig. 3 The role of Notch signaling pathway on maintaining eMSC at a quiescent state. a** The relative proliferative efficiency as determined by a proliferation assay ($n = 7$). **b** The relative gene expression of *Ki67* after activation or inhibition of Notch signaling in eMSC ($n = 6$). **c** Representative histograms for Ki67 and relative percentage of CD140b$^+$CD146$^+$Ki67$^+$ cells after activation or inhibition of Notch signaling in eMSC ($n = 6$). **d** Representative dual parameter fluorescence dot plot after co-staining for Hoechst and Pyronin Y ($n = 5$). **e** The relative proportion of eMSC in different cell cycle phases ($n = 5$). **f** Relative expression of cell cycle-related genes in eMSC after activation or inhibition of Notch signaling ($n = 5–7$). **g** Protocol of in vitro colony forming assay and relative clonogenic activity of eMSC under different conditions after cultured with JAG1 or DAPT at passage 0 ($n = 6$). Results are presented as mean ± SD; *$P < 0.05$; **$P < 0.01$; ***$P < 0.001$. Statistical analysis was performed using One-way ANOVA followed by Tukey's test or Kruskal–Wallis test followed by Dunn's post-test for multiple group comparison. Abbreviation: eMSC endometrial mesenchymal stem-like cells, rh recombinant.

transcription factors, which binds to β-catenin as coactivator[24]. Here, the increase of TCF/LEF transcriptional activity in eMSC upon JAG1 treatment shown by the TCF/LEF luciferase reporter assay confirmed the above observation (Fig. 4a). As expected, proximity ligation assay (PLA) revealed more binding between NICD and active β-catenin in the nuclei of eMSC cultured on JAG1 coated plate than the DMSO control. In contrast, the nuclear PLA positive signal reduced after treatment with DAPT (Fig. 4c).

*Action of WNT activity on Notch related proteins.* Next, we investigated the functional role of canonical WNT signals on the expression of Notch related proteins in eMSC. Recombinant WNT3A and Wnt/β-catenin inhibitor XAV939 were used to activate and inhibit the WNT/β-catenin signaling pathway, respectively (Fig. 4d, e). Recombinant WNT3A remarkably increased the expression of Notch-related proteins NICD, HEY-2, and HES-1 in eMSC (Fig. 4d, f–h). In contrast, XAV939 down-regulated the expression of the Notch downstream molecules (Fig. 4d, f–h).

*Endometrial label retaining stromal cells (LRSC) in a mouse menstrual-like model.* Menstruation occurs in humans and most primates, whereas most placental mammals, including mice go through an estrous cycle. A mouse model of menstruation has been established to recapitulate the menstruation events in human endometrium[25]. In this study, the mouse menstrual-like model was employed to gain insight into the role of Notch activity on stem cells during endometrial regeneration in vivo. The label retaining cells approach was used to identify the endometrial stem/progenitor cells in mice because lack of defined stem cells markers for the mouse endometrium. In the model, sesame oil was injected into one uterine horn of the mice on day 4 of pseudopregnancy to induce decidualization, which resulted in an increase in endometrial thickness and obliteration of the lumen of the treated horn on day 7. The enlargement of the treated horn continued to day 9. During this time course, the color of the horn changed from pink to dark red/purple, indicating readiness of tissue breakdown. The size of the decidualized uterine horn decreased on day 10 and became macroscopically the same as the control horn on day 12 (Supplementary Fig. 4b).

Histological examination showed an open lumen and intact endometrium in the uterine horn before decidualization. Three days after oil injection (day 7), typical features of decidualization including intense vascularization, densely packed decidualized stromal cells and closure of the lumen were observed. Tissue breakdown occurred on day 9, resulting in slough off the decidualized tissue into the lumen. Repair of the endometrium commenced on day 10 when the luminal epithelium re-epithelialized and the stromal cells began to proliferate. The repair was completed by day 12, and the morphology of the repaired endometrium resembled that of the control (Supplementary Fig. 4c, d).

The stem/progenitor cells of mice were labeled with BrdU in prepubertal state. We showed previously a steady percentage of

BrdU$^+$ cells was achieved after a 6-week chase[26]. Hence, the BrdU$^+$ cells after a 6-week chase were referred as LRSC, and they accounted for $2.73 \pm 0.33\%$ ($n = 4$) of the total stromal cells (Fig. 5a, b). The labeled mice were used to prepare the menstrual-like model, decidualization was induced in one horn with the contralateral horn as control. The percentage of BrdU$^+$ stromal cells at decidualization declined rapidly when compared to that of the control (Fig. 5a, b). Only $0.38 \pm 0.05\%$ ($n = 4$) of LRSC were detected in the decidualized uterine horn (Fig. 5a, b and e). The number of LRSC increased at endometrial breakdown but began to decline at early and late repair (Fig. 5a, b). Overall, the percentages of LRSC in the decidualized uteri horn during repair were lower than that of the control (Fig. 5b).

*Endometrial LRSC proliferate during repair.* Proliferation of LRSC was evaluated by co-expression of BrdU and Ki67. No pro-liferating LRSC were found in the endometrium before decid-ualization, suggesting that these cells remained in a quiescent state (Fig. 5c, d). A high percentage of proliferating LRSC was observed ($31.99 \pm 5.20\%$, $n = 4$) in the decidualized tissue. The percentages remained high at tissue breakdown ($22.62 \pm 8.61\%$, $n = 4$). The percentage of proliferating LRSC was highest at early repair ($49.86 \pm 6.09\%$, $n = 4$) and declined rapidly at late repair (Fig. 5c, d, $20.46 \pm 12.49\%$, $n = 4$), which contributed to the decrease in the number of LRSC observed at late repair (Fig. 5b). Interestingly, a small population of LRSC near the endometrial/ myometrial junction did not express Ki67 during decidualization (Fig. 5e). We postulated that these non-proliferating LRSC would be functioning during early and late repair (Fig. 5c).

*Proliferating endometrial LRSC express Notch1.* Since Notch ligands JAG1 and DLL4 were expressed in the mouse endome-trium during tissue repair (Fig. 6a), we evaluated the activation of Notch signal in LRSC. Using an anti-Notch1 antibody that mainly recognizes the activated intracellular domain (ICD) of Notch1[27], we localized Notch1 to the nuclei of LRSC. Before decidualization, $25.0 \pm 5.0\%$ of the LRSC were Notch1$^+$. The percentage increased to $44.88 \pm 44.37\%$ (Fig. 6b, c) in decidualized endometrium. The percentage of Notch1$^+$ LRSC remained rela-tively constant until late repair when it decreased to $30.87 \pm 5.57\%$, which was significantly lower than that in the early repair (Fig. 6b, c). Interestingly, the Notch1$^+$ LRSC proliferated (BrdU$^+$Notch1$^+$Ki67$^+$ cells) only during events of endometrial remodeling i.e., at decidualization ($14.24 \pm 5.0\%$), tissue break-down ($9.16 \pm 2.20\%$) and early repair ($21.39 \pm 17.42\%$, Fig. 6d, e)

*DAPT treatment prolongs endometrial repair after menstrual-like breakdown.* To determine the function of Notch signaling on LRSC, DAPT was injected into the uterine cavity on the day of endometrial breakdown. The relative mRNA expression of *HES-1* and *HEY-2* was significantly decreased in the DAPT treated group than the vehicle group (Fig. 7a, b). Inhibition of the Notch signals in the mouse endometrium was also confirmed by western blotting showing reduction of NICD and HES-1 expression in the

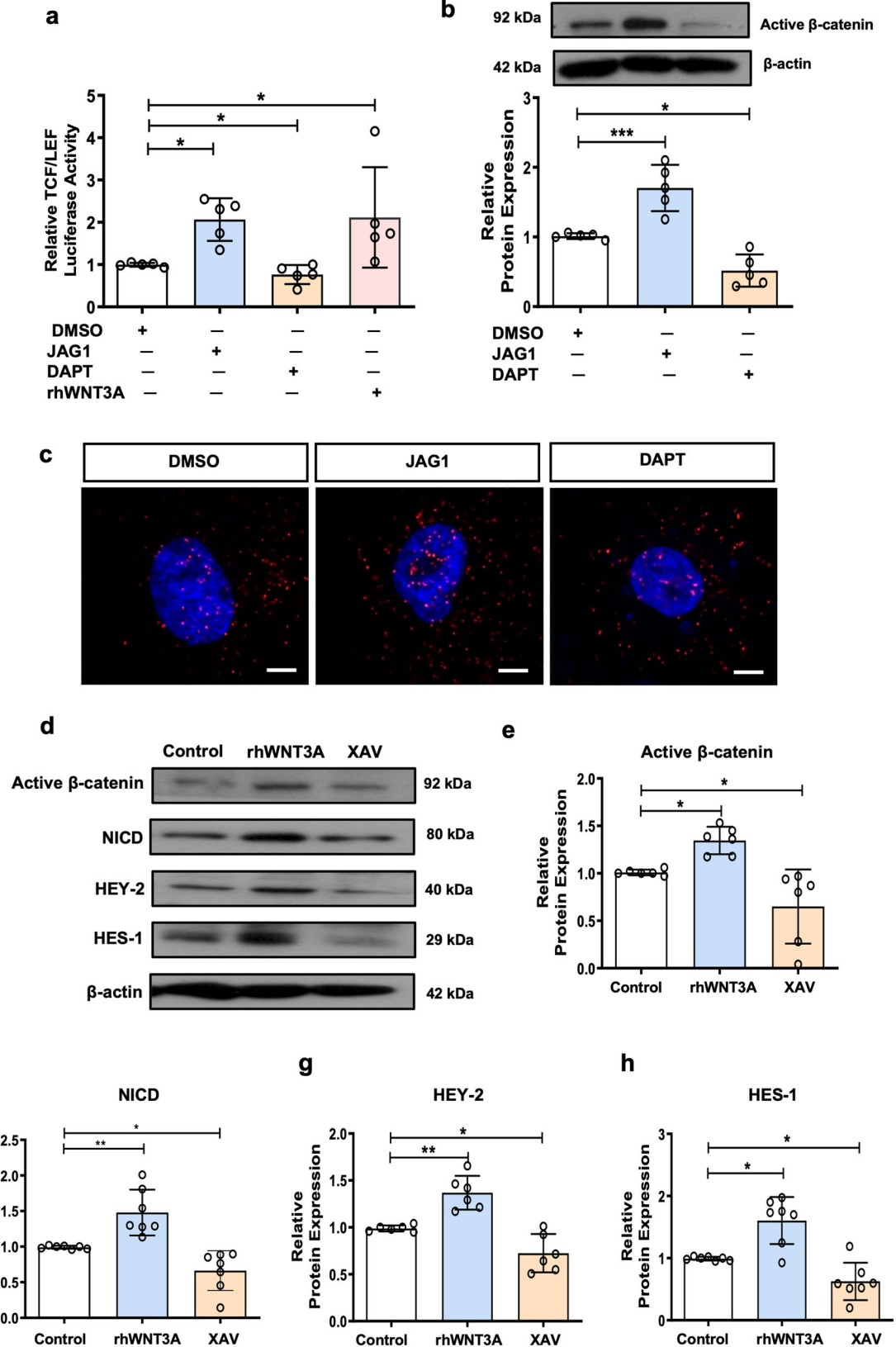

tissue after DAPT treatment (Fig. 7c–e). The crosstalk between Notch and Wnt signaling was also evaluated in vivo. Our results showed that DAPT treatment remarkably reduced the relative mRNA expression of Wnt target genes *AXIN2* and *LEF1* when compared to the vehicle group (Fig. 7f, g). Downregulation of the Wnt signaling was confirmed at protein level by reduction of

active β-catenin expression in the mouse endometrium after DAPT treatment (Fig. 7h, i).

In the vehicle treated group, the stromal compartment was fully restored, and only a small portion of the unrepair luminal epithelium was observed at early repair (Fig. 8a). Endometrial regeneration was completed, and the luminal epithelium became

**Fig. 4 Interaction of Notch and WNT/β-catenin pathway in eMSC. a** The TCF/LEF luciferase signal of eMSC after activation or inhibition of Notch signaling (*n* = 5). **b** Representative western blotting images and semi-quantitative analysis of active β-catenin protein expression on eMSC after Notch signals activation or inhibition (*n* = 5). **c** Representative fluorescence images of in situ proximity ligation assay on eMSC after Notch signals activation or inhibition (*n* = 3), scale bar: 10 μm. **d** Representative western blotting images of active β-catenin and Notch signals protein expression in eMSC after canonical Wnt signaling activation or inhibition (*n* = 6–7). **e–h** Semi-quantitative analysis of expression of active β-catenin and Notch signaling proteins in eMSC after canonical Wnt signaling activation or inhibition (*n* = 6–7). Results are presented as mean ± SD; *$P < 0.05$; **$P < 0.01$; ***$P < 0.001$. Statistical analysis was performed using One-way ANOVA followed by Tukey's test for multiple group comparison. eMSC endometrial mesenchymal stem-like cells, rh recombinant.

intact by late repair (Fig. 8a). When DAPT was administered into the uterine cavity, most part of the luminal epithelium was absent during early repair (Fig. 8a). Even at late repair, sites without luminal epithelium remained, indicating incomplete regeneration of the endometrium (Fig. 8a). Consistent with the morphologic changes, the endometrial thickness of the DAPT treated group was reduced remarkably compared with the vehicle group (Fig. 8b). Taken together, these results suggested that the Notch signaling was required for the post-menstrual regeneration and inhibition of this pathway delayed tissue repair. Moreover, the proportion of proliferating LRSC declined significantly in the DAPT treated group, which resulted in a higher proportion of LRSC detected at both the early and the late repair than the vehicle group (Figs. 8c and d, 9a, b).

Dual immunofluorescence staining revealed that the proportion of Notch1[+] LRSC at early repair was lower in the DAPT treated group than the vehicle group (Fig. 9c, d). The inhibitory effect continued at late repair (Fig. 9c, d). There were also less proliferating Notch1[+] LRSC during the early repair upon DAPT treatment (Fig. 9e, f). The finding explained the presence of more LRSC in the DAPT treated group than the vehicle group (Fig. 9a, b).

## Discussion

Since the identification of eMSC in human endometrium, very few studies have investigated their functional roles in endometrial repair after menstruation[1,28]. The underling mechanism regulating the activities of eMSC remains largely unknown. Here, we demonstrated that Notch signaling plays a role in the maintenance of eMSC. Importantly, inhibition of Notch signaling affected the function of LRSC to repair the endometrium after tissue breakdown in a mouse menstrual-like model.

Notch signaling is essential for homeostasis of somatic stem cells[8,29]. In line with our findings, Spitzer et al. reported an upregulation of Notch signaling molecules in eMSC[14]. Here, we manipulated the Notch signals with pharmacological agonist or antagonist, and showed promotion of eMSC maintenance upon Notch stimulation and vice versa. Similar finding was detected in the SUSD2[+] eMSC[30]. Together, these observations highlight a key role for Notch signals in the maintenance of eMSC.

The Notch family receptors are highly involved with stem cell maintenance and cell fate specification[29,31]. Notch1 is required for the maintenance of mouse adult neural stem/progenitor cells[32]. In bone marrow stromal/stem cells, Notch2 enhances self-renewal and proliferation ex vivo while preserving their osteogenic and chondrogenic differentiation potential[33]. In human endometrium, our results revealed that eMSC expressed Notch1. Knockdown of Notch1 reversed the stimulatory effect of JAG1 on the activity of eMSC, demonstrating the involvement of Notch1 for eMSC functions. Endothelial cells exhibit strong expression of JAG1[14,34]. Since eMSC reside in the perivascular regions of the human endometrium, it will be worth investigating the ligand-receptor interaction between the endothelial cells and the eMSC. We speculate that the Notch signaling in eMSC will be activated by the adjacent endothelial cells to maintain quiescence of the stem cells.

Quiescence and self-renewal are critical for the reservation of somatic stem cells[35]. Stem cells are maintained in a quiescent state to protect the cells against loss of self-renewal potential and rapid exhaustion of the stem cell pool[36]. Here we demonstrated that activation of Notch signaling by JAG1 prevented the proliferation of eMSC and regulated the entrance of eMSC into the cell cycle. Moreover, our data showed that the JAG1 induced quiescent state in eMSC was reversible. These results are in line with the observations in muscle and neural stem cells, where Notch signaling is a critical factor regulating quiescence of the cells[37,38]. We propose that specific signals from niche cells during menstruation stimulate eMSC to re-enter the cell cycle for restoration of the endometrial lining. Several Notch ligands have been identified in human endometrium and their expressions were tightly regulated by steroid hormones in the menstrual cycle[11]. In future studies, it will be important to understand the action of these ligands on eMSC in controlling the balance between the quiescent and the activated state in the menstrual cycle.

Several studies have demonstrated crosstalk between Wnt and Notch signaling in somatic stem cells[39,40]. The two signaling pathways are required to regulate stemness and differentiation of human fallopian tube organoids[41]. In human neural progenitor cells, Notch-target genes mediate the promoting effect of Wnt activation on neurogenesis[39]. We previously demonstrated that myometrial cells regulated the self-renewal of eMSC via WNT/β-catenin signaling[17]. Our current results showed that activation of the Wnt signaling stimulated the quiescent eMSC to proliferate. These observations prompted us to investigate the existence of a crosstalk between the Notch and the Wnt signaling in modulation of eMSC activities. Indeed, our results revealed that inhibition of Notch signaling downregulated the expression of active β-catenin in eMSC while activation of Notch signaling showed the opposite effect. Similar results were observed on the activity of Notch signals when Wnt signaling was pharmacologically manipulated in eMSC. The interaction of these two signaling pathways were further confirmed in vivo. Taken together, our results suggest that the interplay between the Notch and the Wnt signaling is vital in the regulation of eMSC activities in the human endometrium.

Another important finding of our study was that blocking of the Notch signaling had a negative impact on endometrial repair via suppressing the regeneration ability of mouse endometrial stem/progenitor cells. Using the BrdU pulse-chase approach, our group previously characterized LRSC from mouse gestational and postpartum endometrium[26]. In an artificial mouse endometrial breakdown and repair model, the perivascular LRSC proliferate and contribute to a remarkable stromal expansion during endometrial repair[42]. Here, we observed similar findings in our established mouse model. During decidualization, the number of LRSC decreased due to the dynamic changes in the endometrium. Although some LRSC were detected in the decidua tissue, a small population of non-proliferating LRSC were found in the basal region of the endometrium. These LRSC proliferated at endometrial breakdown, indicating that the endometrium can initiate

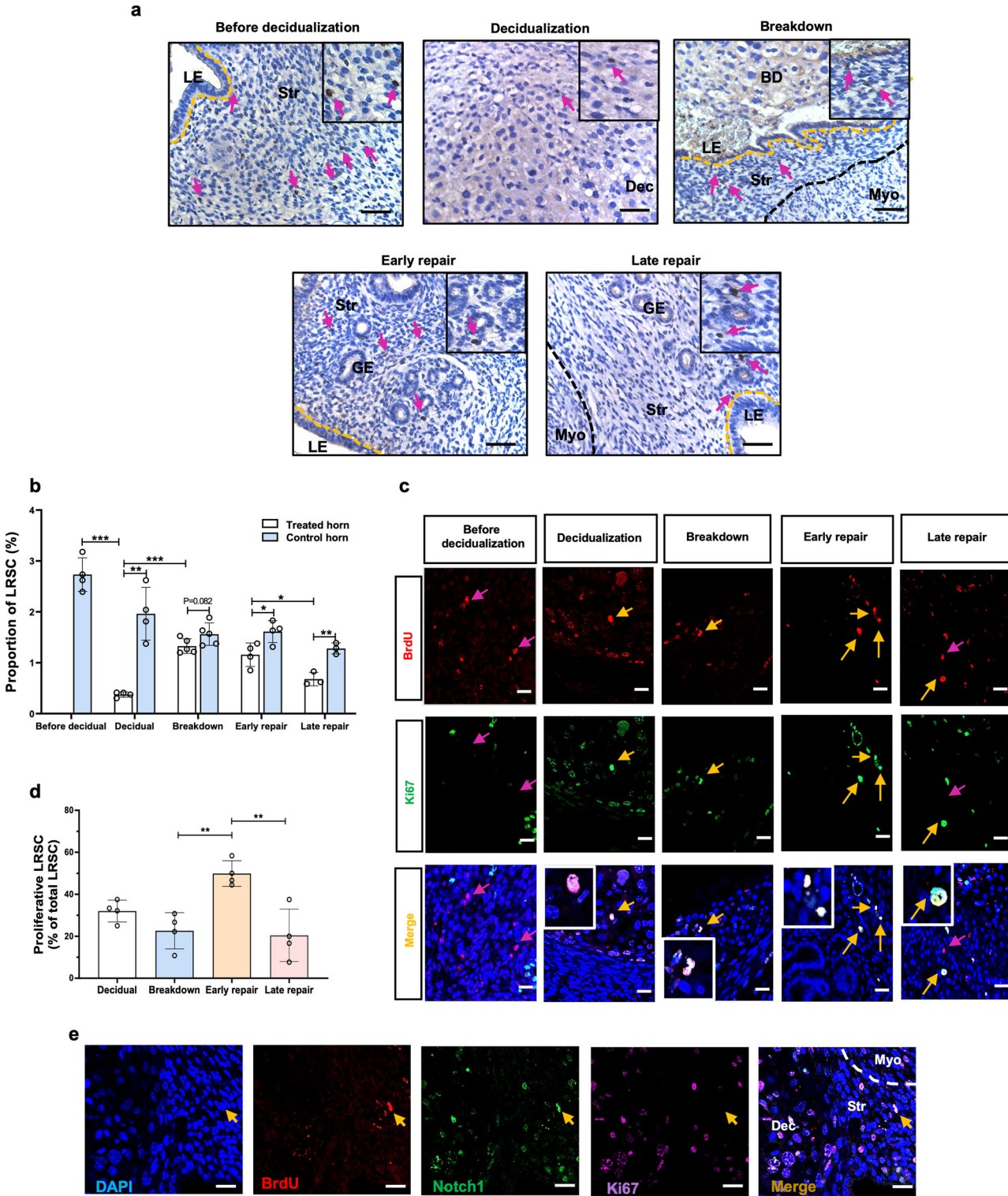

**Fig. 5 Localization of BrdU-labeled stromal cells in the mouse menstrual-like model. a** Localization of LRSC in uterine tissues at different time points in the mouse menstrual-like model. *Inserts* are enlarged images of the BrdU-labeled cells. Red arrows show the distribution of LRSC. **b** Changes in proportion of LRSC during endometrial repair. BrdU-labeled stromal cells are expressed as a percentage of total stromal cells. **c** Representative immunofluorescence images showing LRSC expressing the proliferating marker Ki67 (yellow arrow) at decidualization, breakdown, early and late repair. No Ki67 (red arrow) was observed in LRSC before decidualization. *Inserts* are enlarged images of LRSC co-expressing Ki67, scale bar: 20 μm. **d** The percentage of proliferating LRSC at different time points in the mouse menstrual-like model. **e** LRSC remained in the dense stromal compartment during decidualization. Yellow arrows indicate absence of Ki67 on LRSC, scale bar: 20 μm. Results are presented as mean ± SD; *P < 0.05; **P < 0.01; ***P < 0.001. $n = 3$–5 per group. Statistical analysis was performed using a two-tailed unpaired Student's t test for two group comparison and One-way ANOVA followed by Tukey's test for multiple group comparison. Abbreviation: BrdU bromodeoxyuridine, BD breakdown, Dec decidua, GE glandular epithelium, LE luminal epithelium, LRSC label retaining stromal cells, Myo myometrium, Str stroma.

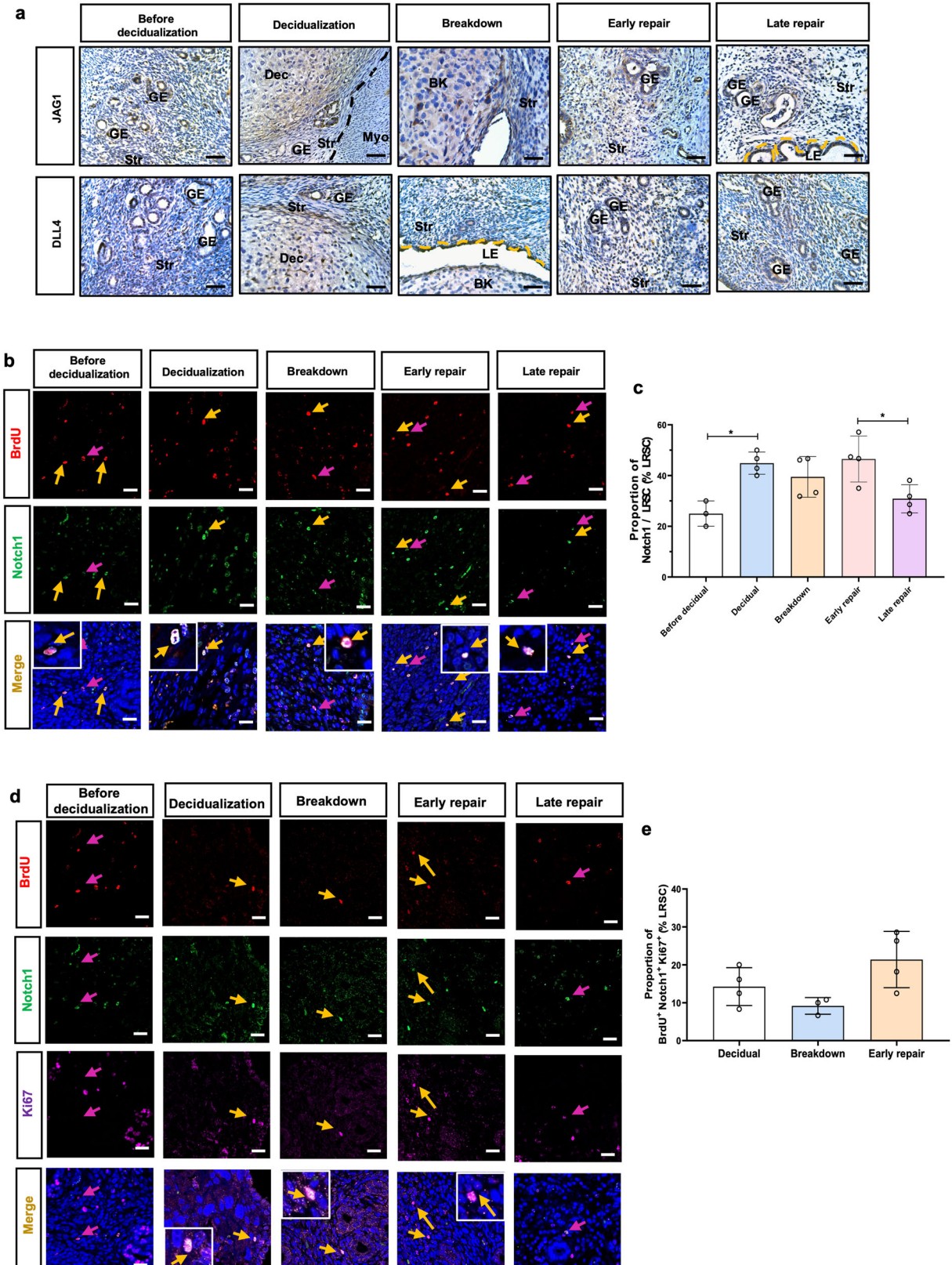

repair at the time of shedding. Findings in the human endometrium support this observation, whereby endometrial regeneration occurs simultaneously with tissue breakdown in a piecemeal fashion[43,44]. The proportion of LRSC in our mouse menstrual-like model was similar to that reported by Kaitu'u-Lino et al.[42].

Notch components are expressed in the mouse endometrium throughout the estrous cycle[45]. Therefore, it was not surprising that DAPT induced dysfunctional endometrial repair in the mouse menstrual-like model. Alteration in stromal cells' proliferative ability may have contributed to size reduction of the

**Fig. 6 Characterization of LRSC in mouse menstrual-like model. a** Representative immunohistochemical images of JAG1 and DLL4 on uterine tissues at different time points. **b** Representative immunofluorescence images showing LRSC expressing Notch1 (yellow arrow) before decidualization, during decidualization, breakdown, early and late repair. Red arrows indicated absence of Notch1 in LRSC. *Inserts* are enlarged images of LRSC expressing Notch1. **c** The percentage of Notch1+ LRSC at different time points in the mouse menstrual-like model. **d** Representative immunofluorescence images show proliferating LRSC expressing Notch1 (yellow arrow) at decidualization, breakdown and early repair. No expression of Notch1 (red arrow) in proliferating LRSC before decidualization and at late repair. *Inserts* are enlarged images of LRSC co-expressing Notch1 and Ki67. **e** The percentage of BrdU +Notch1+Ki67+ at different time points in the mouse menstrual-like model. Data are presented as mean ± SD; *$P < 0.05$. $n = 3$–5 per group. Scale bar: 20 μm. Statistical analysis was performed using One-way ANOVA followed by Tukey's test for multiple group comparison. BrdU bromodeoxyuridine, Dec decidua, GE glandular epithelium, Str stroma. LE luminal epithelium, Myo myometrium, BD breakdown.

repaired endometrium. Endometrial re-epithelialization is the first step in repair of the injured mucosal surfaces[46]. This process was also delayed in our DAPT-treated mice. Before decidualization, ~25.0% of the Notch1+ LRSC were detected and the expression of Notch1 increased thereafter until late repair. The finding is consistent with a pregnant mouse model, whereby the initiation of implantation promotes the expression of Notch1 in the stromal cells[12]. Interestingly, the proliferating Notch1+ LRSC in our model were only observed at decidualization, breakdown, and early repair when the endometrium underwent a dynamic remodeling. Our recent publication demonstrated that during endometrial breakdown and early repair, the LRSC resided in a hypoxic microenvironment[15]. Immunofluorescence staining revealed that the hypoxic-located LRSC underwent proliferation and expressed Notch1, consistent with the involvement of the Notch signaling during these events. More importantly, the Notch-responsive stem cells were activated for tissue repair. Suppression of Notch signaling in the uterine microenvironment reduced the proliferation activity of the Notch1+ LRSC and consequently impaired the repair process. Similar results were observed in the Notch transcription factor *Rbpj* knockout mice, where uterine-specific *Rbpj* knockout led to dysfunctional post-partum uterine repair[47]. The Notch signaling may also alter the inflammatory process during endometrial repair. It has been reported that systemic Notch inhibition prolongs the expression of inflammatory cytokines, resulting in a delayed bone fracture healing[48]. Moreover, suppression of Notch signaling in the postpartum uterus induces an exaggerated immune response and unbalanced macrophages polarization[47]. Therefore, it is important to clarify the role of Notch signaling on the inflammation during menstruation in the near future.

In this study, we found that activation of Notch signaling can better maintain the phenotypic expression of eMSC in vitro. Suppression of Notch signals in vivo delayed endometrial repair after tissue breakdown in the mouse menstrual-like model. Collectively, our data demonstrate that Notch signaling plays an important role in regulating the activity of endometrial stem/progenitor cells in vitro and in vivo. Given the correlation of attenuation Notch signaling with female infertility, understanding the precise mechanisms involved may provide a promising therapeutic approach in the future.

## Materials and methods
**Human tissues.** Full thickness endometrial tissues were collected from 27 women aged 41–52 years (mean age 46.4 years), who underwent abdominal hysterectomy for benign non-endometrial pathologies (Table S1). Only pre-menopausal women with regular menstrual cycle and not taken any hormonal therapy for at least 3 months were recruited for the study. The phase of the endometrium was categorized as proliferative ($n = 16$) and secretory phase ($n = 11$) by an experienced histopathologist, who evaluated the hematoxylin-eosin stained endometrial sections of each sample. A written consent was signed by each patient after detailed counseling prior to participation of the study. Ethical approval was obtained from the Institutional Review Board of The University of Hong Kong/Hospital Authority Hong Kong West Cluster (UW20-465) and The Institutional Review Board of the University of Hong Kong-Shenzhen Hospital ([2018]94).

**Isolation of endometrial stromal cells.** The isolation of single endometrial stromal cells was performed according to our previous study[17]. In brief, endometrial tissue was minced into small pieces and digested with PBS containing collagenase type III (0.3 mg/ml, Worthington Biochemical Corporation, NJ, USA) and deoxyribonuclease type I (40 μg/ml, Worthington Biochemical Corporation) at 37 °C for 1 h. After two rounds of digestion, Ficoll-Paque (GE Healthcare, Uppsala, Sweden) centrifugation and anti-CD45 antibody coated Dynabeads (Invitrogen, Waltham, MA, USA) were sequentially used to remove the red blood cells and the leukocytes, respectively. The stromal cells were then separated from the epithelial cells using anti-CD326 (EpCAM) antibody-coated microbeads (Miltenyi Biotec Inc., San Diego, CA, USA). Next, freshly isolated stromal cells were seeded into 100 mm dishes coated with fibronectin (1 mg/ml, Gibco) and cultured in growth medium (GM) containing 10% FBS (Invitrogen), 1% L-glutamine (Invitrogen), and 1% penicillin-streptomycin (Invitrogen) in DMEM/F-12 (Sigma-Aldrich, St Louis, MA, USA). Stromal cells were cultured in a humidified carbon dioxide incubator at 37 °C. The medium was changed every 7 days until the cells reached 80% confluence.

**Magnetic bead selection for endometrial mesenchymal stem-like cells.** EMSC were obtained by two sequential beadings with magnetic beads coated with anti-CD140b and anti-CD146 antibodies[17]. Firstly, the stromal cells were incubated with the phycoerythrin (PE)-conjugated anti-CD140b antibody (R&D Systems, Minneapolis, MN, USA) for 45 min at 4 °C followed by another 15 min incubation with anti-mouse IgG1 magnetic microbeads (Miltenyi Biotech). The obtained cell suspensions were then loaded onto MS columns (Miltenyi Biotech) with a magnetic field to separate the CD140b+ cells. The isolated CD140b+ stromal cells were cultured for 7–10 days to allow degradation of the microbeads during cell expansion. The cells were then trypsinized and incubated with the anti-CD146 antibody-coated microbeads (Miltenyi Biotec Inc.) for 15 min at 4 °C to obtain the CD140b+CD146+ cells for subsequent experiments. The isolated eMSC showed positive expression for CD140b and CD146[17]. The stromal cells at passage 1–3 were used in this study.

**Flow cytometry.** Multi-color flow cytometry was applied to analyze the co-expression of CD140b and CD146 in endometrial stromal cells. The cells were incubated with PE-conjugated anti-CD140b antibody (2.5 μg/ml, PR7212 clone, Mouse IgG1, R&D Systems) and FITC-conjugated anti-CD146 antibodies (5 μg/ml, OJ79c clone, mouse IgG1; ThermoFisher Scientific) in dark for 45 min at 4 °C. The Fluorescent Minus One (FMO) control was included for each antibody. Following the final washing step, the labeled cells were analyzed by a CytoFlex™ flow cytometer (Beckman Coulter, CA, USA). The cells were selected with electronic gating according to the forward and the side scatter profiles. Data were analyzed by the FlowJo Software (Tree Star Inc).

**Activation/inhibition of Notch or WNT/β catenin signaling pathway.** To activate Notch signaling, 48 well plates were firstly coated with fibronectin for 30 min and then with 50 μg/ml protein G (Invitrogen) overnight, washed with PBS and incubated with recombinant human JAG 1 protein (JAG1, 2 μg/ml, R&D System) for 3 h at room temperature as described[49]. Freshly isolated eMSC ($4 × 10^3$ cells/well) were seeded onto the JAG1 coated plates and cultured in GM for 7 days. To inhibit Notch signaling, eMSC were treated with N-[N-(3,5-difluorophenacetyl- l -alanyl)]-(S)-phenylglycine t-butyl ester (1.25 μM, DAPT, R&D Systems) and cultured in GM for 7 days.

For activation of WNT/β-catenin signaling, recombinant human WNT3A (0.01 μg/ml, R&D Systems) or recombinant human WNT5A (0.01 μg/ml, R&D Systems) was supplemented to eMSC cultured in GM. The WNT inhibitor, XAV939 (10 μM, R&D System) was used to inhibit the Wnt signaling. GM containing DMSO was used as control.

**In vitro colony forming assay.** EMSC were cultured in the presence of JAG1 or DAPT for 7 days to form monolayer. These cells were then trypsinized, reseeded at cloning density (35 cells/cm²) onto 6-well plates, and cultured in GM for 14 days to form colonies. Medium was changed every 7 days. The cloning efficiency was evaluated by the number of colonies divided by the number of seeded cells multiplied by 100.

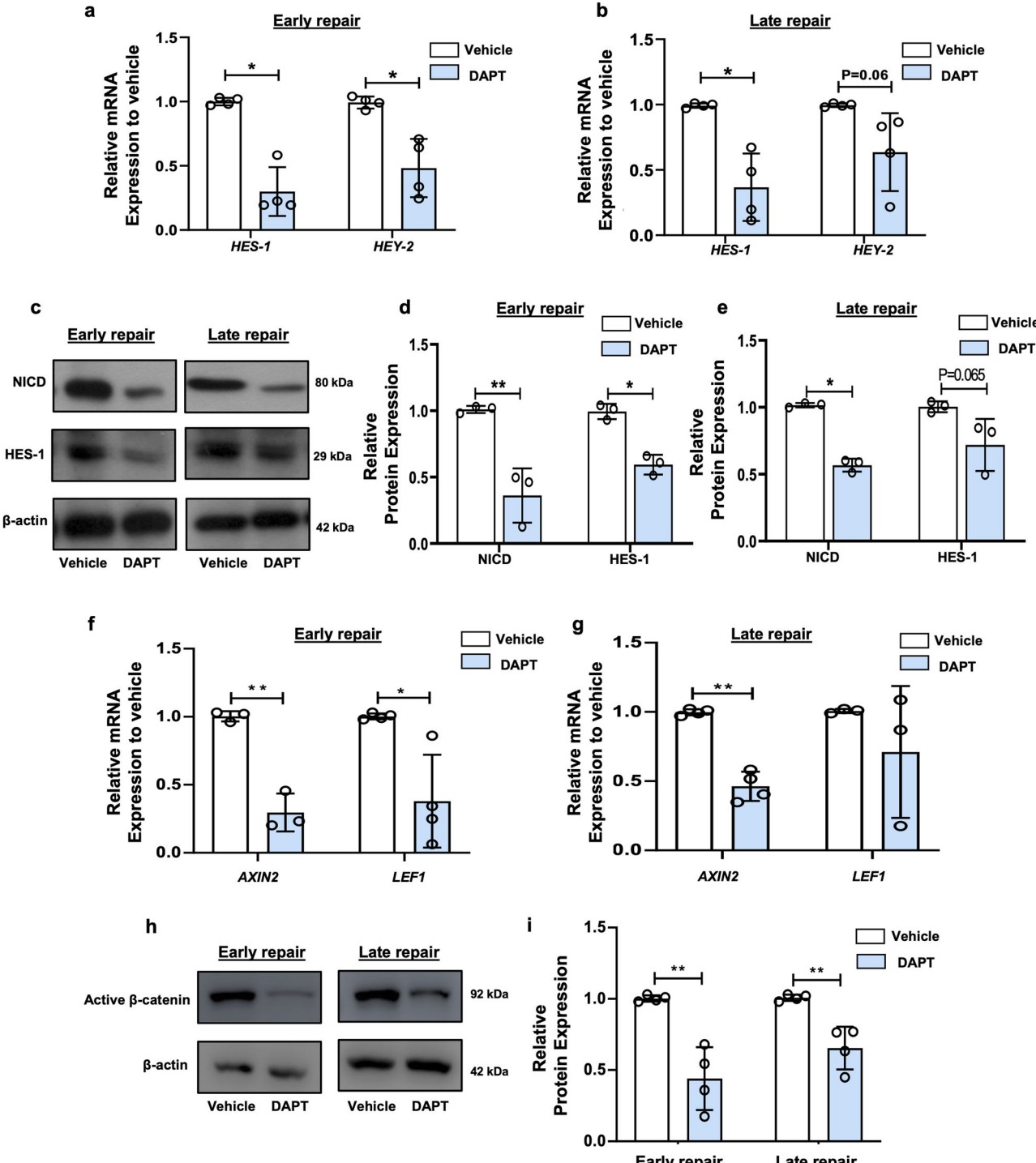

**Fig. 7 The activity of Notch signaling in mouse uterus after DAPT treatment. a** Relative expression of Notch target genes in mouse uterine tissue after DAPT treatment at early repair ($n = 4$). **b** Relative expression of Notch target genes in mouse uterine tissue after DAPT treatment at late repair ($n = 4$). **c** Representative western blotting images of Notch signals protein expression on mouse uterine tissue after DAPT treatment at different time points. **d** Semi-quantitative analysis of expression of Notch signaling proteins in mouse uterine tissue after DAPT treatment at early repair ($n = 3$). **e** Semi-quantitative analysis of expression of Notch signaling proteins in mouse uterine tissue after DAPT treatment at late repair ($n = 3$). **f** Relative expression of Wnt target gene on mouse uterine tissue after DAPT treatment at early repair ($n = 3$–4). **g** Relative expression of Wnt target genes in mouse uterine tissue after DAPT treatment at late repair ($n = 3$–4). **h** Representative western blotting images of active β-catenin expression in mouse uterine tissue after DAPT treatment at different time points. **i** Semi-quantitative analysis of active β-catenin expression in mouse uterine tissue after DAPT treatment at different time points ($n = 4$). Results are presented as mean ± SD; *$P < 0.05$; **$P < 0.01$. Statistical analysis was performed using a two-tailed unpaired Student's t test.

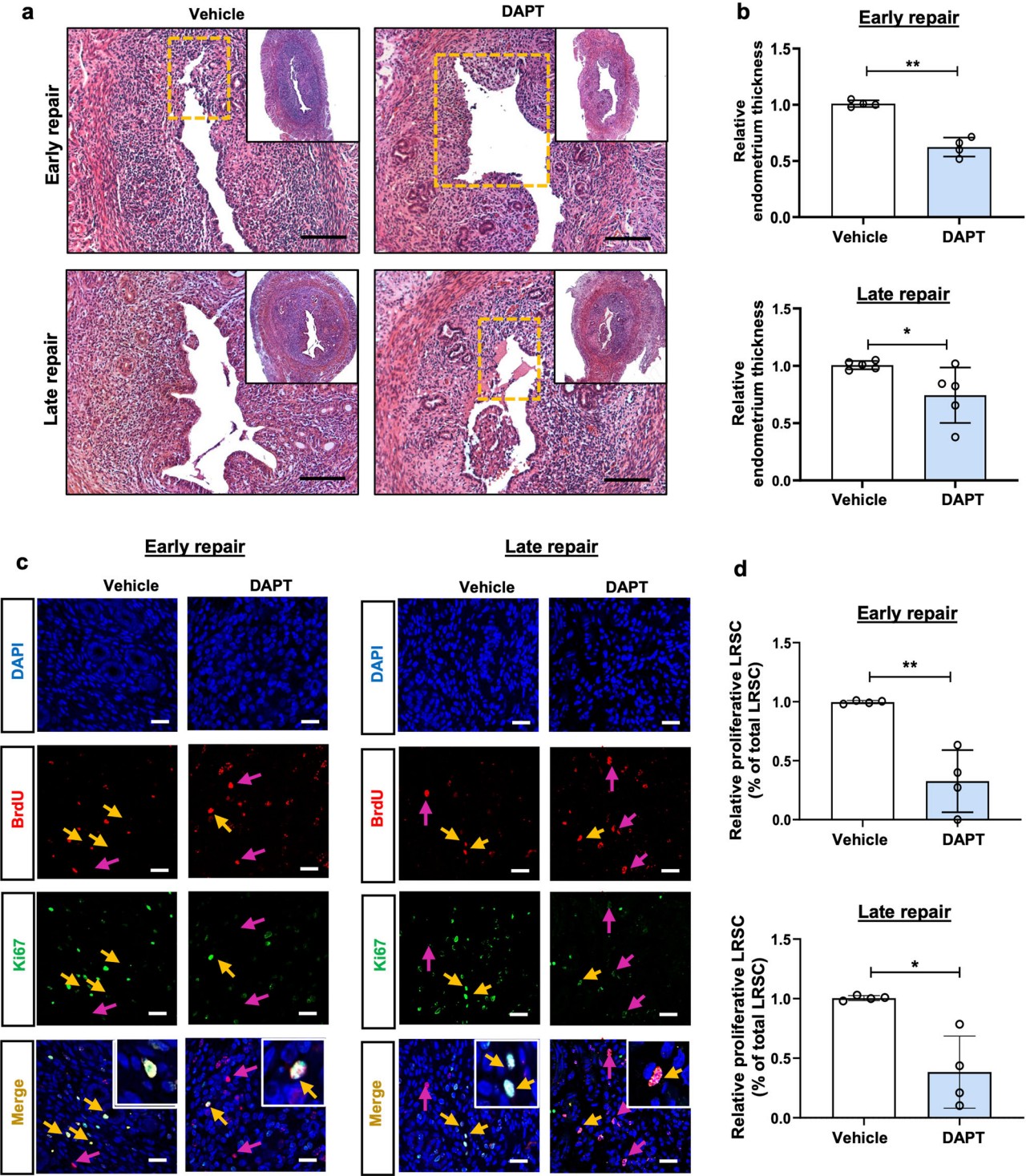

**Fig. 8 Proliferation of LRSC after DAPT treatment. a** Histological analysis of hematoxylin and eosin-stained uteri after DAPT treatment at early and late repair. *Inserts* show the whole transverse section of the uterine tissue. Dashed boxes indicate unrepair sites. Scale bar: 50 μm. **b** Relative endometrium thickness after DAPT treatment at early and late repair. **c** Representative immunofluorescence images of proliferating LRSC after DAPT treatment at different time points. Yellow arrows show LRSC expressing the proliferating marker Ki67. Red arrows indicate absence of Ki67 on LRSC. *Inserts* are enlarged figures of LRSC expressing Ki67. Scale bar: 20 μm. **d** Relative percentage of proliferating LRSC after DAPT treatment at different time points. Results are presented as mean ± SD; *$P < 0.05$; **$P < 0.01$. $n = 3$–5 per group. Statistical analysis was performed using a two-tailed unpaired Student's t test. BrdU, bromodeoxyuridine.

**Cell proliferation assay**. The CyQUANT™ NF Cell Proliferation kit (Thermo Scientific) was used to determine the proliferative ability of eMSC. In brief, eMSC ($1 \times 10^3$ cells/well) were seeded into 96-well plates and cultured in GM. The cells were either treated with JAG1 or DAPT for 3 days. Cells treated with DMSO were used as control. After washing with PBS, 100 μl of dye binding solution was added to each well and cultured for 1 h at 37 °C. Fluorescence intensity was measured by a fluorescence microplate reader with excitation at 485 nm and emission at 530 nm.

**G0/G1 cell cycle analysis**. G0/G1 cell cycle analysis was performed as described[50]. Endometrial MSC were treated with or without JAG1 or DAPT for 7 days. Then

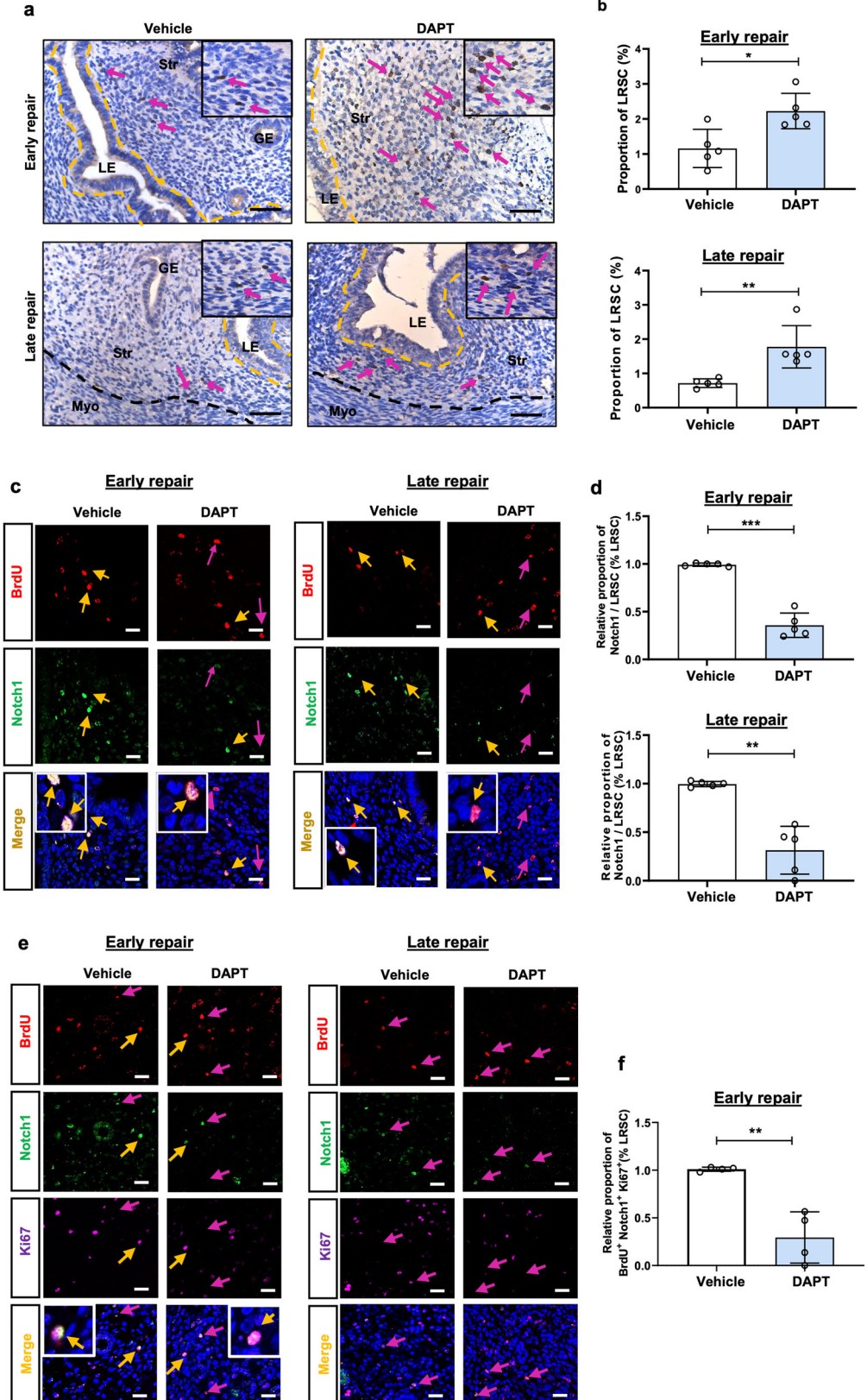

cells were trypsinized and incubated with APC-conjugated anti-CD140b antibody (2.5 μg/ml, PR7212 clone, Mouse IgG1, R&D Systems) and FITC-conjugated anti-CD146 antibodies (5 μg/ml, OJ79c clone, mouse IgG1; ThermoFisher Scientific) in dark for 45 min at 4 °C. The cells were then incubated with Hoechst dye (1 μg/ml) for 30 min at 37 °C. The Pyronin Y dye (1 μg/ml) was directly added to the cells and incubated for 30 min at 37 °C and analyzed using a Fortessa flow cytometer

(BD Biosciences) at the Imaging and Flow cytometry Core, Center for PanorOmic Sciences (CPOS), LKS Faculty of Medicine, The University of Hong Kong.

**WNT reporter assay.** EMSC ($2–5 × 10^4$ cells/well) were seeded into 24-well plates and cultured to 80% confluence when they were co-transfected with 4 μg of TOP flash or FOP flash vector and 1 μg of PRL-TK (Renilla-TK-luciferase vector,

**Fig. 9 DAPT treatment increases the proportion of LRSC at early and late repair. a** Localization of LRSC in uterine tissues after DAPT treatment at different time points. *Inserts* are enlarged images of BrdU-labeled cells. Red arrows show the distribution of LRSC in the tissues. **b** Proportion of LRSC after DAPT treatment at different time points. **c** Representative immunofluorescence images showing LRSC expressing Notch1 (yellow arrow) after DAPT treatment at early and late repair. Red arrows indicate absence of Notch1 on LRSC. *Inserts* are enlarged images of LRSC expressing Notch1. **d** Relative percentage of Notch1+ LRSC after DAPT treatment at different time points. **e** Representative immunofluorescence images showing proliferating LRSC expressing Notch1 (yellow arrow) after DAPT treatment at early repair and late repair. Red arrows indicate the absence of Notch1 in proliferating LRSC. *Inserts* are enlarged images of LRSC co-expressing Notch1 and Ki67. **f** Relative percentage of BrdU+Notch1+Ki67+ cells after DAPT treatment at early repair. Results are presented as mean ± SD; *$P < 0.05$; **$P < 0.01$; ***$P < 0.001$. $n = 3–5$ per group. Scale bar: 20 μm. Statistical analysis was performed using a two-tailed unpaired Student's t test. BrdU bromodeoxyuridine, GE glandular epithelium, LE luminal epithelium, Myo myometrium, Str stroma.

Promega, Madison, WI, USA) using Lipofectamine 2000 (Invitrogen). Subsequently, the cells were treated with JAG1 or DAPT for 48 h. Some cells were treated 0.01 μg/ml recombinant human WNT3A (R&D Systems) as positive control. The luciferase activity was measured using a GLOMAX™ 96 microplate luminometer. Transfection efficiency was determined by firefly luciferase activity normalized against the Renilla luciferase activity. The TOP/FOP ratio was presented as a measure of the TCF/LEF transcription.

**Fluorescence protein labeling assay.** Human recombinant JAG1 (R&D System) was fluorescently labeled with Alexa Fluor™ 488 Microscale Protein Labeling Kit (Thermo Scientific) according to the manufacturer's instructions. Briefly, JAG1 was incubated with 50 μL of sodium bicarbonate solution and Alexa Fluor dye for 1 h at room temperature. Then the protein-dye conjugate was flowed through a gel separation column (Bio-rad Biogel P-30) to purify the labeled protein, which was aliquoted and stored at −80 °C for subsequent studies.

**Gene silencing.** Endometrial MSC ($2 × 10^4$/well) were seeded onto 24-well plates, treated with JAG1 and cultured in OptiMEM medium (Invitrogen). After overnight incubation, the cells were transfected with 10 pmol of siRNA directed against Notch 1 (ID s9633 and s9635; Ambion) or random siRNA with scrambled sequence (Ambion) using the Lipofectamine RNAiMax transfection reagent (Invitrogen) according to the manufacturer's instructions. The medium was changed to GM at 24 h post-transfection and the cells were trypsinized for flow cytometry analysis. The knockdown efficiency was determined by western blotting analysis (Fig. 2e).

**Immunofluorescence staining of cells.** For immunofluorescence staining, ~8000 cells were cytospun at 12,000 rpm for 10 min and fixed with 4% paraformaldehyde for 10 min. Permeabilization was conducted with 0.1% Triton-X 100 for 10 min, followed by blocking with 5% BSA for 30 min. Primary antibodies (Table S2) or isotype-matched control antibodies were incubated overnight at 4 °C. The next day, the corresponding secondary antibody (Table S3) was added and incubated for 1 h. The cell nuclei were detected by staining with DAPI (Thermo Scientific) and mounted with mounting medium (Dako). The slides were washed with PBST between steps and all the steps were conducted at room temperature unless specified. Multi-spectrum fluorescence images were captured using a LSM 710 inverted confocal microscope and a LSM ZEN 2010 software (Carl Zeiss, Munich, Germany) at the CPOS, The University of Hong Kong. The total cell number and number of triple positive (CD140b+CD146+NICD+) cells were counted. At least 500 cells were counted from each sample.

**Western blot analysis.** Cultured eMSC or mouse endometrial tissues were lysed in cell lysis buffer (Ambion, Grandisland, NY, USA) in the presence of protease inhibitors. Then 5 μg of denatured protein samples were separated on 10% SDS-PAGE and transferred to polyvinylidene difluoride membranes (Immobilon™-P, Milllipore). After blocking with 5% skim milk for 1 h at room temperature, the membranes were incubated with appropriate primary antibodies overnight at 4 °C followed by horseradish peroxidase conjugated secondary antibodies for 1 h at room temperature (Table S4). The protein expression was detected by the Western Bright ECL Kit (Advansta, CA, USA). The intensities of the western blot bands were quantified by the Quantity One software and normalized to that of β-actin. Uncropped scans of western blots are shown in Supplementary Figs. 5 and 6.

**Quantitative real-time polymerase chain reaction.** Quantitative real-time polymerase chain reaction (qPCR) with Taqman probes were used (Table S5). The total RNA was isolated using the Absolutely RNA microprep kit (Agilent Technologies, Santa Clara, CA, USA) according to the manufacturer's instructions. The concentration of total RNA was quantified by spectrophotometry. RNA was reversed transcribed to cDNA by the PrimeScript DNA Reverse Transcription kit (Takara, Japan). PCR was conducted by a 7500 Real-Time PCR System (Applied Biosystems). The mixtures were incubated at 50 °C for 2 min and 95 °C for 10 min, followed by 40 cycles of 15 s at 95 °C and 1 min at 60 °C. Gene expression was measured in triplicate and presented as relative gene expression using the $2^{-\Delta\Delta Ct}$ method and normalized to 18S as internal control.

**In situ proximity ligation assay (PLA).** The in-situ interaction of NICD and active β-catenin was determined using the Duolink™ II secondary antibodies and detection kits (Sigma–Aldrich, #DUO92001, #DUO92005 and #DUO92008) according to the manufacturer's instructions. First, fixed cells were incubated with the PLA probes and primary antibodies against NICD and β-catenin (Table S2) overnight at 4 °C. The Duolink™ secondary antibodies were added in the following day and incubated at 37 °C for 1 h. The secondary antibodies were ligated together to form a closed circle by the Duolink™ ligation solution when the antibodies were in proximity (<40 nm) to each other. Polymerase and amplification buffer were then applied to amplify the positive signal (red dot) of the resulting closed circles and the cells were visualized with a LSM 800 inverted confocal microscope and a LSM ZEN 2010 software (Carl Zeiss, Munich, Germany) at the CPOS, The University of Hong Kong. The total number of cells and positive signals were counted. At least 500 cells were counted from each sample.

**Animal and housing condition.** Mice were provided by Center of Comparative Medicine Research at The University of Hong Kong. All experimental procedures performed in this study were approved by the Committee on Use of Live Animals in Teaching and Research, The University of Hong Kong, Hong Kong. The mice were kept under standard conditions with a light/dark cycle of 12 h/12 h and free access to food and water.

**Animal study design.** The experimental setup is shown in Supplementary Fig. 4a. Day 19 prepubertal C57BL/6J female mice were labeled with BrdU according to our previous study[26]. In brief, the mice were intraperitoneally injected with BrdU (50 mg/g of body weight; Sigma Aldrich) twice daily for four consecutive days and allowed to grow without further labeling. After a 6-week chase, the standard protocol to induce endometrial breakdown and repair was performed[25]. In brief, female mice were mated with vasectomized >6-week-old C57BL/6J male mice (day 0). Pseudopregnant BrdU labeled female mice were identified by the presence of a vaginal plug on the next day (day 1). On day 4 of pseudopregnancy, 30 μl of sesame oil was injected into the left uterine horn to induce decidualization while the right horn was not treated as control. The mice were euthanized, and their uteri were harvested on day 4 of pseudopregnancy (before decidualization), decidualization (day 7), breakdown (day 9), early repair (day 10) and late repair (day 12).

For the Notch inhibition study, decidualization was induced in both uterine horns on day 4 of pseudopregnancy. When endometrial breakdown occurred on day 9, DAPT (10 mg/kg) was injected into one uterine horn. This dosing regimen was sufficient to achieve desired Notch ablation without adversely affecting the animal's health[51]. The other uterine horn received the same volume of saline and served as a control. The uteri were collected on day 10 and day 12. The harvested tissues were fixed overnight with 4% paraformaldehyde and embedded in paraffin blocks for immunohistochemistry and immunofluorescent staining.

**Histological analysis of mouse endometrial thickness.** Paraffin sections (5 μm) were stained with hematoxylin (Sigma-Aldrich) and eosin (Sigma-Aldrich) using standard protocols. Average endometrial thickness was measured from transverse section of the uterine horns–the vertical distance from the luminal epithelium to the endometrial–myometrium interface using the Image-Pro Plus software (version 6.0, Media Cybernetics) from 10 serial sections of the same animal.

**BrdU immunohistochemistry staining of mouse endometrial tissues.** BrdU immunohistochemistry was performed as described[26]. Paraffin sections were dewaxed and underwent antigen retrieval, followed by denaturation with 0.1 N HCl for 45 min. The sections were then quenched with 3% hydrogen peroxide for 10 min, blocked with 5% BSA/PBS for 1 h, and incubated with sheep anti-BrdU antibody (1:500 dilution; Abcam) or isotype control at 4 °C overnight. On the next day, the sections were incubated with donkey anti-sheep biotinylated secondary antibodies (1:400 dilution; Abcam) for 1 h and then with the Vectastain ABC reagent (Vector Laboratories) for 30 min. BrdU positive staining was revealed by DAB solution (Dako) under a Zeiss Axioskop II microscope (Carl Zeiss). The sections were counterstained with the Mayer's hematoxylin, dehydrated, and

mounted using aqueous mounting medium (Dako). Images were acquired using a Photometrics CoolSNAP digital camera (Roper Scientific).

**Dual and triple immunofluorescence staining of mouse endometrial tissues**. Paraffin sections underwent deparaffinization, rehydration, antigen denaturation and blocking as described above. For dual immunofluorescence staining, the two primary antibodies (Table S2) were co-incubated at 4 °C overnight. The slides were then incubated with the corresponding secondary antibodies (Table S3) at 37 °C for 1 h. For triple staining, the anti-BrdU antibody staining was conducted first, followed by incubation with the other two primary antibodies at 4 °C overnight and subsequently the corresponding secondary antibodies on the next day. The slides were stained with DAPI (Thermo Scientific) and mounted with a fluorescence mounting medium (Dako). Multi-spectrum fluorescence images were captured by a Carl Zeiss LSM 800 inverted confocal microscope and the Zeiss LSM ZEN 2019 software (Carl Zeiss) at the CPOS, The University of Hong Kong.

**Enumeration of BrdU-Labeled cells**. BrdU-labeled cells were counted in a blinded manner as described[26]. One transverse and one longitudinal section from each animal of different time points were analyzed. Images of the entire mouse uterine horn was acquired by a digital camera (Roper Scientific). The number of BrdU$^+$ cells and the total number of cells in the stromal compartment were counted using the ImageJ software (NIH Image; National Institutes of Health). At least 2000 nuclei per uterine horn per mouse at each time point were counted. We only considered whole nuclei stained BrdU$^+$ cells as LRSC. The percentage of BrdU$^+$ cells were determined by dividing the number of BrdU$^+$ cells by the total number of nuclei counted in each section.

**Statistics and reproducibility**. Data were analyzed using the GraphPad PRISM software (version 8.00; GraphPad Software Inc., San Diego, CA, USA). Distribution normality was tested using the Shapiro-Wilk test. Differences between two groups were analyzed using the Mann–Whitney U test for non-parametric data and the two-tailed unpaired Student's t test for parametric data. One-way ANOVA followed by Tukey's test or Kruskal–Wallis test followed by Dunn's test were used for multiple group comparison. The number of samples for statistical tests can be found in the figure legends where applicable. Data are represented as mean ± SD. A difference with P-value of <0.05 is considered as significant.

**Reporting summary**. Further information on research design is available in the Nature Research Reporting Summary linked to this article.

## Data availability
The data that support the findings of this study are available in Supplementary Data and from the corresponding authors upon reasonable request.

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

## Acknowledgements

We are grateful to all the women who agreed to donate their tissue samples for this study. We sincerely acknowledge Joyce Yuen, our project nurse and Stella Wang, our research assistant. To all gynecologists at Queen Mary Hospital and The University of Hong Kong Shenzhen Hospital for the collection of the samples. We are also grateful to the staffs at the Center for PanorOmic Science (CPOS), Imaging and Flow cytometry Core and Center of Comparative Medicine Research, The University of Hong Kong for their technical assistance in this study. This study was supported by funding from the National Natural Science Foundation of China/Research Grants Council Joint Research Scheme (N_HKU 732/20), Science, Technology & Innovation Commission of Shenzhen Muni-cipality (JCYJ20180508153031), and The Hong Kong University Shenzhen Hospital Scientific Research Training Plan (HKUSZH20192003).

## Author contributions

S.Z.: participated in most of the experimental work, analysis of data and writing of the manuscript. R.W.S.C.: contributed to the study concept, writing, and editing of the manuscript. E.H.Y.N.: contributed to the recruitment of patients and sample collection for this study. W.S.B.Y.: contributed to the study design, critical discussion, and proof reading of the manuscript.

## Competing interests

All authors declare no competing interests.
