## [Peer Review File · Communications Biology]

Reviewers' comments:

Reviewer #1 (Remarks to the Author):

GENERAL COMMENT:

It is, in general, a well-written molecular study about the role of Notch and Wnt signaling pathways in the function and regulation of endometrial mesenchymal stem cells. The authors properly establish several in vitro assays to study the activation and deprivation of the Notch pathway and the correlation with the Wnt route. Finally, the animal model helps elucidate how the endometrial mesenchymal stem cells are involved in endometrial repair after menstruation.

Introduction

-At the end of this section, the authors perfectly define the objectives of their work. However, they also anticipate some results of the work presented ("We showed that activation of Notch signaling can better maintain the phenotypic expression of eMSC", "Our results illustrated that suppression of Notch signals delayed the endometrial repair after breakdown. In conclusion, we demonstrate that Notch signaling plays an important role in regulating the activity of endometrial stem/progenitor cells in vitro and in vivo.") that I would suggest moving to the discussion part.

Material and methods

-Why did the authors opt for CD140b and CD146 stem cell markers and not others?

-Why was not the DAPT condition included in the colony-forming assay (as the authors did with the proliferation assay)?

-I suggest talking earlier about Flow cytometry, as eMSC isolation was one of the initial steps of the work plan.

-I would suggest not including results in this section (p34, line 586).

-Did the authors consider establishing a KO animal for any gene in the Notch and/or Wnt signaling pathways?

Discussion

-I suggest being more specific in the final conclusion, in the sentence "our data demonstrate the importance of Notch signaling in regulating the activity of endometrial stem/progenitor cells in vitro and in vivo".

Reviewer #2 (Remarks to the Author):

This study investigated the role of Notch signaling in maintaining endometrial mesenchymal stem cells in the endometrium by using primary cell culture and the mouse menstrual-like model. Overall, the manuscript is interesting but there are some additional points that need to be addressed before further consideration.

Comments

1. As stated in Lines 46-47, the human endometrial mesenchymal stem like cells (eMSC) are the population of cells co-expressing perivascular markers CD140b and CD146. However, the description on lines 94-96 does not match details provided in Fig 1E and Fig 2G and is therefore misleading. Did the authors use eMSC or simply endometrial stromal cells in the experiments described in this section?

2. Line 247, please give a brief explanation of the method to identify label retaining cells on lines 726-728.

3. The mouse menstrual-like model was treated with DAPT and the activity of Notch signaling was confirmed by showing inhibition in the mouse uterus. However, although DAPT treatment could inhibit LRSC proliferation, it is just one of the potential mechanisms that delays the endometrial repair. The other potential mechanisms reported in the literature should be addressed in the Discussion.

4. Only the in vitro evidence regarding the interaction between Notch signaling with WNT/beta-catenin signaling is shown. It is suggested to add the in vivo results from the mouse menstrual-like model that were treated with DAPT.

5. The methods section is redundant and could be shortened with the rest of the details provided in the supplement.

6. The contents in Lines 74-84 might be not appropriate in Introduction. The sentences in Lines

466-468 and lines 480-482 need to be revised.

Reviewer #3 (Remarks to the Author):

In this manuscript, the authors first investigate the role of Notch signaling in endometrial mesenchymal stromal/stem-like cells (eMSC) first by activating or inhibiting Notch signaling in eMSC culture. They identify effects on cell proliferation and cell cycle progression or quiescence. They further demonstrate reversibility through interaction with the Wnt signaling. For in vivo validation, they use a mouse model mimicking the menstrual cycle and show that endometrial BrdU label-retaining stromal cells proliferate during endometrial repair in a Notch1 signaling-dependent mechanism. Notch1 inhibition delayed post-menstruation endometrial repair. This is an interesting study that highlights the potential role of the Notch1 signaling in regulating eMSC proliferation and endometrial repair.

There are two major concerns:

1. The study was conducted with appropriate procedures that are well detailed. The manuscript is well structured and easy to follow. However, it clearly needs in-depth editing by a native English-speaker.

2. Data comparing eMSC and unfractionated stromal cells is not convincing, especially in Fig. 1 A-D and Fig. 2A. Results in these subfigures are quite similar between the two cell populations, except for a few values (3 max) that are much higher in the eMSC population. Do the authors have an explanation for this heterogeneity in the eMSC population? Unfortunately, these are the first results presented in the manuscript and they penalize an otherwise well-performed study.

Minor comments:

- Line 96: Please briefly present JAG1 for non-specialist readers.
- Lines 123-125: The authors state that "Our results showed a non-significant higher expression level of NOTCH1, NOTCH2 and NOTCH3 mRNA between eMSC and unfractionated stromal cells (Fig 2A)". As mentioned above, levels are globally similar, with the exception of a few values. Please rephrase and comment.
- Line 197: "between different cell cycle phases" (in Fig 3 legend)
- Line 216: Please explain TCF/LEF for non-specialist readers (binds beta-catenin as coactivator)
- Line 226: Please present XAV939 as a Wnt/beta-catenin inhibitor, here rather than in line 230
- Lines 248-249: This is not a sentence
- Line 435-437: The authors state that "The high level of endogenous Notch related genes expressed by eMSC when compared to unfractionated stromal cells suggest the involvement of Notch signaling pathway in stem cell regulation". Please remove this sentence since the figure is not convincing.
- Lines 502-503: This is not a sentence
- Line 635: Briefly describe and give source of TOP flash and FOP flash vectors

29th July 2022

Dear Communications Biology Editorial Team,

RE: Manuscript COMMSBIO – 22-1368-T “The Role of Notch Signaling in Endometrial Mesenchymal Stromal/Stem-like Cells Maintenance”.

Thank you for your email dated 9th June 2022 containing the reviewer’s reports and advising us on the outcome of the captioned manuscript. We thank the reviewers for their comments and suggestions. We have addressed all the points raised as detailed below and incorporated the changes (highlighted text) in the revised manuscript

Reviewer #1:

Introduction

1. At the end of this section, the authors perfectly define the objectives of their work. However, they also anticipate some results of the work presented (“We showed that activation of Notch signaling can better maintain the phenotypic expression of eMSC”, “Our results illustrated that suppression of Notch signals delayed the endometrial repair after breakdown. In conclusion, we demonstrate that Notch signaling plays an important role in regulating the activity of endometrial stem/progenitor cells in vitro and in vivo.”) that I would suggest moving to the discussion part.

Response: *Following the reviewer’s suggestion, we have moved this section into the discussion part and reorganize to make the final conclusion more specific (line 73-79, line 536-543).*

Material and methods

2. Why did the authors opt for CD140b and CD146 stem cell markers and not others?

Response: *There are two sets of surface markers commonly used to enrich for endometrial mesenchymal stem-like cells. One is the co-expression of CD140b and CD146, which was chosen for this study. The other is the single marker - SUSD2. Our group previously compared the expression of both sets of markers in human endometrial samples. We found the percentage of CD140b+CD146+ cells were less fluctuating than that of the SUSD2+ cells among different human endometrial samples. Since the endometrium is also a very dynamic tissue with cyclical changes, we selected these two markers to isolate eMSC to reduce sample variability.*

3. Why was not the DAPT condition included in the colony-forming assay (as the authors did with the proliferation assay)?

Response: *We have performed this experiment and included the result in the revised manuscript as Fig 3G (line 182-184).*

4. I suggest talking earlier about Flow cytometry, as eMSC isolation was one of the initial steps of the work plan.

Response: *We have rearranged this method accordingly (line 591-600).*

5. I would suggest not including results in this section (p34, line 586).

Response: *The sentence has been removed.*

6. Did the authors consider establishing a KO animal for any gene in the Notch and/or Wnt signaling pathways?

Response: *As mentioned in our animal experimental setup, decidualization is an essential part in the establishment of the mouse menstrual-like model. However, many studies have indicated that uterine-specific Notch-knockout mice exhibited significant decidualization defect. In addition, conventional Wnt-knockout mice showed abnormal development of mouse uterus. Therefore, we used DAPT, a chemical Notch signaling inhibitor, to study the functional role of Notch signals in endometrial regeneration in vivo.*

Discussion

7. I suggest being more specific in the final conclusion, in the sentence “our data demonstrate the importance of Notch signaling in regulating the activity of endometrial stem/progenitor cells in vitro and in vivo”.

Response: *We have revised the final conclusion (line 536-543).*

Reviewer #2:

1. As stated in Lines 46-47, the human endometrial mesenchymal stem like cells (eMSC) are the population of cells co-expressing perivascular markers CD140b and CD146. However, the description on lines 94-96 does not match details provided in Fig 1E and Fig 2G and is therefore misleading. Did the authors use eMSC or simply endometrial stromal cells in the experiments described in this section?

Response: *We used eMSC in the experiments in this section. When eMSC were cultured in the growth medium without growth factors and niche signals (the control group) these cells underwent spontaneously differentiation and gradually lost their stemness, as indicated by the decrease in percentage of CD140b⁺CD146⁺ cells using flow analysis. Therefore, after culturing eMSC in the presence of Notch activator or inhibitor, we performed flow cytometry to determine which condition could better maintain the phenotypic expression of CD140b & CD146. The figure legend for Fig 1E (line 104-106) and Fig 2G (line 141-143) have been revised indicating that eMSC were used in these experiments.*

2. Line 247, please give a brief explanation of the method to identify label retaining cells on lines 726-728.

Response: *We have included a brief description of the protocol in the method (line 682-684).*

3. The mouse menstrual-like model was treated with DAPT and the activity of Notch signaling was confirmed by showing inhibition in the mouse uterus. However, although DAPT treatment could inhibit LRSC proliferation, it is just one of the potential mechanisms that delays the endometrial repair. The other potential mechanisms reported in the literature should be addressed in the Discussion.

Response: *Indeed, other potential mechanisms may also delay endometrial repair, we have revised the discussion and included other published reports (line 527-533).*

4. Only the in vitro evidence regarding the interaction between Notch signaling with WNT/beta-catenin signaling is shown. It is suggested to add the in vivo results from the mouse menstrual-like model that were treated with DAPT.

Response: *We have included these results as Figure 7F-I in the revised manuscript (line 356-361, Figure 7F-7I).*

5. The methods section is redundant and could be shortened with the rest of the details provided in the supplement.

Response: *We have shortened the methods and described the details in the supplement.*

6. The contents in Lines 74-84 might be not appropriate in Introduction. The sentences in Lines 466-468 and lines 480-482 need to be revised.

Response: *We have rewritten the introduction paragraph and revised the mentioned sentences (line 470-472, line 483-485).*

Reviewer #3:

1. The study was conducted with appropriate procedures that are well detailed. The manuscript is well structured and easy to follow. However, it clearly needs in-depth editing by a native English-speaker.

Response: *We have invited a native English speaker to edit our manuscript.*

2. Data comparing eMSC and unfractionated stromal cells is not convincing, especially in Fig. 1 A-D and Fig. 2A. Results in these subfigures are quite similar between the two cell populations,

except for a few values (3 max) that are much higher in the eMSC population. Do the authors have an explanation for this heterogeneity in the eMSC population? Unfortunately, these are the first results presented in the manuscript and they penalize an otherwise well-performed study.

Response: *Since the endometrium is a dynamic tissue, patient variation together with stage of the menstrual phase may contribute to the heterogeneity in the eMSC population. In the revised Fig1A-D and Fig 2A, we used the Δ CT values to show the distribution of the samples in each population (line 85-77, line 103-104, line 118-119, line 131-132, Fig 1A-1D, Fig 2A).*

Minor comments:

3. Line 96: Please briefly present JAG1 for non-specialist readers.

Response: *We have briefly described JAG1 in the manuscript when it was first presented in the manuscript (line 89-91).*

4. Lines 123-125: The authors state that “Our results showed a non-significant higher expression level of NOTCH1, NOTCH2 and NOTCH3 mRNA between eMSC and unfractionated stromal cells (Fig 2A)”. As mentioned above, levels are globally similar, with the exception of a few values. Please rephrase and comment.

Response: *The high sample variability of using primary endometrial samples is likely to have contributed to the globally similar expression levels of NOTCH1, NOTCH2 and NOTCH3. Since this manuscript focus on the role of Notch1 receptor, the results of Notch2 and Notch3 have been removed for a more concise presentation of the data and the sentence has been revised (line 118-119, line 131-132, Fig 2A).*

5. Line 197: “between different cell cycle phases” (in Fig 3 legend)

Response: *Revised (line 196-197).*

6. Line 216: Please explain TCF/LEF for non-specialist readers (binds beta-catenin as coactivator)

Response: *We have included the explanation as suggested (line 215-219).*

7. Line 226: Please present XAV939 as a Wnt/beta-catenin inhibitor, here rather than in line 230

Response: *This has been updated as suggested (line 226-228).*

8. Lines 248-249: This is not a sentence

Response: *The sentence has been revised (line 249-251).*

9. Line 435-437: The authors state that “The high level of endogenous Notch related genes expressed by eMSC when compared to unfractionated stromal cells suggest the involvement of Notch signaling pathway in stem cell regulation”. Please remove this sentence since the figure is not convincing.

Response: *This sentence has been removed.*

10. Lines 502-503: This is not a sentence

Response: *Revised (line 507-518).*

11. Line 635: Briefly describe and give source of TOP flash and FOP flash vectors

Response: *The details have been included (line 632-634).*

Yours sincerely,

Dr Rachel Chan
Department of Obstetrics &Gynaecology
The University of Hong Kong

REVIEWERS' COMMENTS:

Reviewer #3 (Remarks to the Author):

The authors made changes as requested by this reviewer. However, the modified sentence in lines 85-86 is incorrect since the word "more" is lacking before "abundantly expressed".

Since a revision will be necessary for (at least) this correction, it can be suggested to the authors to present their graphs with deltaCt values on an inverted axis, i.e. with lower deltaCt values on top (Fig. 1A-1D and 2A). Indeed, lower deltaCt values represent higher expression. This is not a request, this is just an info left to the authors' choice.

(Please note that Fig. 3 was not included in the PDF but was only available in the Word document.)

20th September 2022

Dear Communications Biology Editorial Team,

RE: Manuscript COMMSBIO – 22-1368A “*The Role of Notch Signaling in Endometrial Mesenchymal Stromal/Stem-like Cells Maintenance*”.

Thank you for your email dated 13th September 2022 containing the reviewer's report and advising us on the outcome of the captioned manuscript. We thank the reviewers for their comments and suggestions. We have addressed all the points raised as detailed below and incorporated the changes in the revised manuscript

Reviewer #3:

1. To present the graphs with deltaCt values on an inverted axis, i.e. with lower deltaCt values on top (Fig. 1A-1D and 2A). Indeed, lower deltaCt values represent higher expression.

Response: Following the reviewer's suggestion, we revised Fig 1A-D and Fig 2A with an inverted axis for the delta Ct values.

Yours sincerely,

Dr Rachel Chan
Department of Obstetrics &Gynaecology
The University of Hong Kong